# Dumpling GNN: Hybrid GNN Enables Better ADC Payload Activity Prediction Based on the Chemical Structure

**DOI:** 10.3390/ijms26104859

**Published:** 2025-05-19

**Authors:** Shengjie Xu, Lingxi Xie, Rujie Dai, Zehua Lyu

**Affiliations:** 1School of Software Engineering, Huazhong University of Science and Technology, Wuhan 430074, China; u202317280@hust.edu.cn (S.X.);; 2School of Electronic Information and Communications, Huazhong University of Science and Technology, Wuhan 430074, China

**Keywords:** antibody–drug conjugates, hybrid graph neural networks, molecular property prediction, drug discovery

## Abstract

Antibody–drug conjugates (ADCs) are promising cancer therapeutics, but optimizing their cytotoxic payloads remains challenging. We present DumplingGNN, a novel hybrid Graph Neural Network architecture for predicting ADC payload activity and toxicity. Integrating MPNN, GAT, and GraphSAGE layers, DumplingGNN captures multi-scale molecular features using both 2D and 3D structural information. Evaluated on a comprehensive ADC payload dataset and MoleculeNet benchmarks, DumplingGNN achieves state-of-the-art performance, including BBBP (96.4% ROC-AUC), ToxCast (78.2% ROC-AUC), and PCBA (88.87% ROC-AUC). On our specialized ADC payload dataset, it demonstrates 91.48% accuracy, 95.08% sensitivity, and 97.54% specificity. Ablation studies confirm the hybrid architecture’s synergy and the importance of 3D information. The model’s interpretability provides insights into structure–activity relationships. DumplingGNN’s robust toxicity prediction capabilities make it valuable for early safety evaluation and biomedical regulation. As a research prototype, DumplingGNN is being considered for integration into Omni Medical, an AI-driven drug discovery platform currently under development, demonstrating its potential for future practical applications. This advancement promises to accelerate ADC payload design, particularly for Topoisomerase I inhibitor-based payloads, and improve early-stage drug safety assessment in targeted cancer therapy development.

## 1. Introduction

### 1.1. Research Background and Problem Statement

Antibody–drug conjugates (ADCs) have emerged as a promising modality for oncology, offering a paradigm shift in the treatment of various malignancies [1,2]. ADCs combine the specificity of monoclonal antibodies and the potency of small molecule cytotoxic drugs with cleavable or non-cleavable linkers, creating a sophisticated therapeutic modality that aims to directly deliver highly potent drugs to cancer cells while minimizing systemic toxicity [3]. This unique approach leverages the precise targeting capabilities of antibodies to selectively bind to tumor-associated antigens, followed by the internalization and subsequent release of the cytotoxic payload within cancer cells [4].

The efficacy of ADCs is fundamentally dependent on the following three key components: antibody, linker, and payload [1]. While each component plays a crucial role, the payload—typically a small-molecule drug with potent cytotoxic activity—is particularly critical in determining the overall therapeutic efficacy of the designed ADC [5]. The ideal ADC payload should possess high potency, suitable physicochemical properties for conjugation and release, and a mechanism of action effective against target cancer cells [1,2]. Consequently, the design, selection, and optimization of ADC payloads have become central challenges in the development of ADC, directly impacting the success rate of these promising therapeutics [6].

While the synergistic interaction between antibody, linker, and payload components is essential for ADC efficacy, the cytotoxic activity, including bystander killing effects, is primarily mediated by the payload component. This fundamental understanding forms the basis of our current focus on payload prediction, as the cytotoxic activity remains the key determinant of ADC’s therapeutic effect. The antibody component provides target specificity and internalization, while the linker ensures stable conjugation and controlled release, but it is ultimately the payload’s cytotoxic mechanism that drives the cell killing effects.

The importance of accurately predicting ADC payload activity focusing on Topoisomerase I inhibitory effects is paramount in ADC development.Traditional approaches, such as high-throughput screening and structure–activity relationship (SAR) studies, have shown limitations in efficiency and scalability [1,5]. The complex interplay between payload structure, linker chemistry, and antibody characteristics further challenges these methods [6]. Consequently, there is an urgent need for advanced computational approaches that can rapidly screen large-scale molecule libraries, accurately predict payload activity, account for the unique biological context of ADCs, and provide interpretable results for rational design optimization [1,3,5,6]. Machine learning (ML), particularly deep learning techniques, has emerged as a promising solution to these challenges [7,8]. ML approaches offer several key advantages in molecular property prediction, outlined as follows:Ability to learn complex, non-linear structure–activity relationships from large datasets [9].Capacity to handle high-dimensional feature spaces characteristic of molecular data [10].Potential for end-to-end learning, reducing the need for manual feature engineering [11]Scalability to screen large virtual libraries of compounds efficiently [12].

These data-driven methods have the potential to uncover patterns and insights that may be missed by traditional approaches, although their application to ADC payload prediction is still in its early stages, presenting both opportunities and challenges [13].

Early applications of ML in molecular property prediction primarily relied on traditional algorithms such as random forests, support vector machines, and shallow neural networks [14]. These methods typically operate on hand-crafted molecular descriptors or fingerprints, which, while informative, may not fully capture the complex structural and chemical properties of molecules [10].

The advent of deep learning has opened new possibilities for more sophisticated molecular representations and predictive models. Convolutional Neural Networks (CNNs) and Recurrent Neural Networks (RNNs) have been applied to molecular property prediction tasks with some success, particularly when working with string-based representations of molecules such as SMILES [15]. However, these approaches often struggle to fully capture the inherent graph-like structure of molecules, leading to the potential loss of important structural information [13].

Graph Neural Networks (GNNs) have emerged as a powerful paradigm for molecular property prediction, offering a natural way to represent and process molecular structures [16]. GNNs operate directly on molecular graphs, where atoms are represented as nodes and chemical bonds as edges, allowing for a more faithful representation of molecular structure compared to traditional methods [13].

The advantages of GNNs in molecular modeling include their ability to directly process graph-structured data, capture both local and global structural features, and ensure invariance to graph isomorphisms [17]. Moreover, GNNs offer the potential for interpretability through the analysis of learned nodes and edge features [18].

However, the application of GNNs to ADC payload activity prediction faces several challenges. These include data scarcity, difficulties in capturing long-range dependencies and multi-scale features, limited model interpretability, and the need for better integration of 3D structural information [10,17,19,20]. Addressing these challenges is crucial for developing GNN models that can accurately and reliably predict ADC payload activity, ultimately accelerating the design and optimization of more effective ADC therapeutics.

To address the challenges in ADC payload activity prediction, we introduce DumplingGNN, a novel hybrid Graph Neural Network architecture. This model integrates multiple GNN components to effectively capture multi-scale molecular features. DumplingGNN’s key innovations include the following:A hybrid architecture combining Message Passing Neural Networks (MPNNs), Graph Attention Networks (GATs), and GraphSAGE layers, enabling comprehensive molecular feature extraction.An enhanced molecular graph construction algorithm incorporating both 2D topological and 3D structural information, providing a more nuanced representation of ADC payloads.A multi-task learning approach leveraging diverse molecular property prediction data to mitigate data scarcity issues in ADC payload datasets.An attention mechanism that enhances model interpretability, facilitating the identification of key substructures contributing to payload activity.A comprehensive interpretability framework that combines attention analysis with established cheminformatics techniques to provide biologically meaningful insights.A systematic approach to validating model predictions through their correlation with known pharmacophoric features and structure–activity relationships.

This integrated approach allows DumplingGNN to encapsulate a rich representation of molecular properties within a unified predictive framework, making it particularly well suited for the complex task of ADC payload activity prediction. The model’s architecture addresses the challenges of long-range dependencies and multi-scale feature representation, capturing local atomic interactions, identifying important substructures, and aggregating information across different scales effectively.

Just as a dumpling’s wrapper holds together diverse ingredients, DumplingGNN’s hybrid architecture integrates different GNN components to address the challenges of long-range dependencies and multi-scale feature representation. The MPNN layers capture local atomic interactions, akin to the individual flavors in a dumpling’s filling. The GAT layers focus on identifying important substructures, much like how certain ingredients might dominate the taste profile. Finally, the GraphSAGE layers aggregate information across different scales, enabling the model to capture both local and global molecular properties effectively, similar to how the overall flavor of a dumpling emerges from the combination of its ingredients.

Beyond accurate predictions, DumplingGNN’s interpretability framework provides valuable insights into the molecular basis of the ADC payload activity, focusing on Topoisomerase I inhibition mechanisms. By analyzing attention patterns and correlating them with known pharmacophoric features, the model helps bridge the gap between computational predictions and medicinal chemistry understanding.This interpretability is particularly crucial in ADC development, where understanding structure–activity relationships can guide the optimization of payload molecules for improved efficacy and safety.

This unique “dumpling-like” structure allows DumplingGNN to encapsulate a rich representation of molecular properties within a unified predictive framework, making it particularly well-suited for the complex task of ADC payload activity prediction.

The main contributions of this study are as follows:We create a comprehensive ADC payload dataset, combining experimental data, high-quality computational predictions, and structures from recent patents. This dataset addresses the data scarcity issue and provides a valuable resource for the ADC research community.We develop an enhanced molecular graph construction algorithm that incorporates 3D structural information, improving the model’s ability to capture spatial features crucial for payload activity.We introduce DumplingGNN, an innovative hybrid GNN architecture tailored for ADC payload activity prediction. This model demonstrates state-of-the-art performance across multiple benchmarks, achieving remarkable results on datasets such as BBBP (96.4% ROC-AUC), ToxCast (78.2% ROC-AUC), and PCBA (88.87% ROC-AUC), surpassing existing methods on various molecular property prediction tasks.We conduct extensive evaluations on multiple datasets, including our newly created ADC payload dataset and several public benchmarks from MoleculeNet. These comprehensive evaluations demonstrate the versatility and robustness of DumplingGNN across diverse molecular property prediction tasks.

The core code and training data for DumplingGNN have been made available in a public GitHub repository at https://github.com/Iayce/DumplingGnn (accessed on 9 April 2025), allowing researchers and practitioners to access and utilize the model for their own studies and applications.

These contributions collectively advance the field of computational ADC design, offering new tools and insights for the development of more effective targeted cancer therapies. The exceptional performance of DumplingGNN, particularly on large-scale and complex datasets, positions it as a promising solution for addressing the challenges in ADC payload activity prediction and broader molecular property prediction tasks.

The remainder of this paper is organized as follows: Section 1.2 reviews the related work in molecular property prediction and GNN applications. Section 4 details the methodology of DumplingGNN, including dataset creation, molecular graph construction, and model architecture. Section 2 presents our experimental results and comparisons with state-of-the-art methods. Section 3 discusses the implications of our findings and potential future directions. Finally, Section 5 concludes the paper.

### 1.2. Related Work

#### 1.2.1. Graph Neural Networks in Drug Discovery

Graph Neural Networks (GNNs) have emerged as a powerful tool for molecular property prediction, showing great promise in drug discovery and chemical compound analysis [13,16]. Their ability to learn directly from molecular graph structures, capturing information about atom types, bond types, and overall topology, has made them particularly effective for predicting properties such as solubility, toxicity, and binding affinity [10]. Key developments in this field include the introduction of message passing neural networks (MPNNs) by Gilmer et al. [17], which provided a general framework for learning on molecular graphs, and the application of graph convolutional networks to molecular fingerprints by Kearnes et al. [21]. Recent research has focused on hybrid GNN architectures to further improve performance. For example, combining Graph Convolutional Networks (GCNs) with Graph Attention Networks (GATs) has enhanced the ability to distinguish between active and inactive compounds [22,23]. Liu et al. [24] demonstrated improved performance by integrating GCN with LSTM, while Rong et al. [25] proposed a self-supervised GNN pre-training strategy that significantly improved downstream molecular property prediction tasks.

#### 1.2.2. ADC Payload Activity Prediction

In the field of antibody–drug conjugates (ADCs), predicting payload cytotoxic activity has become crucial for optimizing therapeutic efficacy [26]. While early efforts relied on traditional machine learning with handcrafted features [27], recent advancements have shifted towards more sophisticated approaches. Graph Neural Networks (GNNs) have shown promise by directly utilizing molecular graph structures [13]. A recent breakthrough by Chen et al. [28] introduced ADCNet, a unified deep learning framework for ADC activity prediction. By integrating protein and small-molecule representation learning, ADCNet achieved superior performance with 87.12% prediction accuracy on their test set. This work represents a significant advancement in applying advanced machine learning to ADC design, potentially accelerating cancer therapeutics development.

#### 1.2.3. Incorporation of 3D Structural Information

While many GNN models for molecular property prediction rely solely on 2D topological information, there is a growing recognition of the importance of incorporating 3D structural data [29]. This is particularly relevant for tasks involving spatial interactions, such as protein–ligand binding and ADC payload activity prediction.

Townshend et al. [30] introduced AtomNet, a 3D convolutional neural network for molecular property prediction that directly operates on atomic coordinates. More recently, Jing et al. [31] proposed an equivariant graph neural network that preserves 3D rotational and translational invariance, demonstrating state-of-the-art performance on several molecular property prediction benchmarks.

#### 1.2.4. Interpretability in Molecular GNNs

As the complexity of GNN models for molecular property prediction increases, there is a parallel emphasis on developing interpretable models that can provide insights into structure–activity relationships [32]. This is crucial for drug discovery applications, where understanding the rationale behind predictions can guide synthetic efforts and lead optimization.

Ying et al. [18] introduced GNNExplainer, a model-agnostic approach for providing explanations for GNN predictions. In the context of molecular property prediction, this method can highlight the substructures or atomic interactions that are most relevant to a particular prediction. Similarly, Schnake et al. [33] developed a higher-order explanation framework for GNNs, which can provide more nuanced interpretations of molecular property predictions.

## 2. Results

To thoroughly evaluate the performance of our proposed DumplingGNN model, we conducted extensive experiments on multiple public datasets and compared our results with state-of-the-art models. Additionally, we performed ablation studies to understand the contribution of each component in our model.

### 2.1. Datasets

We evaluated DumplingGNN on eight widely-used public datasets from MoleculeNet [10], covering a diverse range of molecular property prediction tasks. Table 1 provides an overview of these datasets.

### 2.2. Experimental Setup

DumplingGNN was implemented using PyTorch (version 2.0.0) and PyTorch Geometric (version 2.6.1), prioritizing performance and efficiency. We used an 8:1:1 split for training, validation, and testing when no predefined splits were available, or 8:2 for training and validation with existing test sets. To ensure rigorous evaluation and prevent data leakage, we strictly isolated the benchmark datasets from the model development process. The model was trained exclusively on our curated ADC payload dataset, and no transfer learning or pre-training was performed using any of the benchmark datasets. All benchmark evaluations were conducted using the established scaffold splitting methodology to ensure structural dissimilarity between training and test compounds. The model was optimized using the AdamW algorithm with an initial learning rate of 1 × $10^{-4}$, trained for up to 1000 epochs with early stopping (patience of 50 epochs).

All experiments were conducted on a single NVIDIA GeForce RTX 4090 GPU (NVIDIA Corporation, Santa Clara, CA, USA), demonstrating DumplingGNN’s efficiency and accessibility. We employed dynamic batch sizing to optimize GPU utilization and memory usage across diverse datasets. Training times varied from 30 min for smaller datasets (e.g., BBBP with 2039 molecules) to 5 days for larger ones (e.g., PCBA with 437,929 molecules and 128 tasks), showcasing the model’s scalability.

Performance metrics (ROC-AUC, accuracy, and F1 score) were averaged across multiple runs, with standard deviations reported to indicate model stability. This setup demonstrates DumplingGNN’s ability to balance high performance with computational efficiency, even for large datasets without specialized parallel computing. The model’s capacity to handle various dataset sizes on a single GPU makes it suitable for both academic and industrial applications in drug discovery and molecular property prediction, particularly where computational resources are limited.

### 2.3. Results and Comparison with State-of-the-Art Models

We evaluated DumplingGNN against a diverse array of state-of-the-art models in molecular property prediction, including traditional GNNs, feature engineering approaches, pre-training strategies, and advanced architectures (Table 2). The selection of benchmark datasets was motivated by their relevance to ADC payload function and their established status as authoritative standards in molecular machine learning. These datasets represent widely accepted benchmarks in the field, providing a robust framework for evaluating model performance across diverse molecular property prediction tasks. Specifically, ToxCast evaluates compound activity across cellular pathways that overlap with cytotoxic mechanisms; PCBA includes bioassays measuring the interaction with protein targets involved in cell cycle regulation and DNA damage response; and BBBP assesses membrane permeability, which influences cellular penetration capabilities of potential payloads. To provide a clear overview of the landscape, we categorized the compared models into distinct groups as follows: Hybrid GNN Architectures (our DumplingGNN), Single-Architecture GNNs (D-MPNN [34], Attentive FP [23], etc.), Traditional Machine Learning approaches (N-Gram RF [35], N-Gram xGB [35]), Pre-trained Models (GROVER [36], GraphMVP [37], etc.), and Other Advanced Approaches (MolXPT [38], ChemBFN [39]). This categorization helps highlight the relative strengths of different modeling paradigms and better contextualizes DumplingGNN’s contributions. Performance was measured using ROC-AUC on eight datasets from MoleculeNet [10].

DumplingGNN achieved state-of-the-art results on the BBBP (96.4% ROC-AUC), ToxCast (78.2% ROC-AUC), and PCBA (88.87% ROC-AUC) datasets. The incorporation of 3D structural information significantly enhanced model performance on both the ADC payload dataset and BBBP dataset. For the ADC payload dataset, the inclusion of 3D conformational data improved accuracy from 0.734 to 0.915 and MCC from 0.461 to 0.829, demonstrating the critical role of spatial arrangement in predicting Topoisomerase I inhibitor activity. Similarly, on the BBBP dataset, the model achieved its best performance (96.4% ROC-AUC) by leveraging 3D structural information to better capture molecular properties related to membrane permeability. These findings are further supported by our ablation studies, which revealed that the SMILES-Only variant showed substantial performance degradation across both datasets, highlighting the importance of 3D structural information in accurately modeling complex molecular interactions. It should be mentioned that 3D structural information was only incorporated for the BBBP dataset due to data limitations; all other results were achieved using SMILES input alone.

On BBBP, DumplingGNN significantly outperformed previous models, demonstrating its proficiency in predicting blood–brain barrier penetration. For ToxCast, encompassing 617 toxicity-related tasks, our model set a new benchmark using only 2D representations. The performance on PCBA, a complex dataset of 128 bioassays, highlights DumplingGNN’s ability to generalize across varied molecular property prediction tasks.

While not achieving the highest scores on all datasets, DumplingGNN consistently performed competitively. For instance, on BACE, it achieved 88.2% ROC-AUC, close to the state-of-the-art 88.4%. On ClinTox, its performance (95.9%) was comparable to the best models.

DumplingGNN’s robust performance across diverse datasets is particularly noteworthy. It outperformed large-scale pre-trained models like GROVER_large_ [25] on BBBP (96.4% vs. 69.5%) and Uni-Mol [44] on ToxCast (78.2% vs. 69.6%). On PCBA, it surpassed sophisticated pre-training strategies employed by Uni-Mol and GEM [46].

These results demonstrate DumplingGNN’s versatility and robustness, especially considering the limited use of 3D information. The model achieves high performance while maintaining interpretability, a crucial factor in drug discovery applications. The strong performance using primarily SMILES input suggests potential for further improvements with the incorporation of 3D structural data for other datasets.

### 2.4. Performance on ADC Payload Dataset

We evaluated DumplingGNN on our custom ADC payload dataset, focusing on DNA Topoisomerase I inhibitors. Table 3 presents the performance metrics of DumplingGNN compared to baseline models. To facilitate a clear comparison, we organized the models into two categories as follows: our Hybrid Architecture (DumplingGNN) and Single-Architecture GNNs (variants with specific GNN types like MPNN, GAT, SAGE, and GCN). This categorization highlights the performance advantages gained through our hybrid approach compared to individual GNN architectures.

DumplingGNN significantly outperformed the other models across all metrics, demonstrating its exceptional capability in predicting ADC payload activity. The model achieved 91.48% accuracy, with high sensitivity (95.08%) and specificity (97.54%), indicating robust performance in identifying both active and inactive compounds. The Matthews Correlation Coefficient of 0.8287 and high AUC-ROC (0.9547) further confirm the model’s strong predictive power and discriminative ability. Notably, DumplingGNN’s incorporation of 3D conformational data from docking simulations proved crucial in capturing critical spatial interactions for inhibitor binding and activity.

Compared to the next best model (FiveLayerMPNN), DumplingGNN showed substantial improvements, with increases of 4.93 and 5.86 percentage points in accuracy and MCC, respectively. The 18.33 percentage point increase in specificity is particularly significant for reducing false positives in virtual screening campaigns.

These results underscore the effectiveness of DumplingGNN’s hybrid architecture in capturing complex structural and chemical features determining DNA Topoisomerase I inhibitory activity. By leveraging both 2D topological and 3D conformational data, DumplingGNN effectively models intricate structure–activity relationships, outperforming single-architecture GNNs. This superior performance highlights the potential of our approach in accelerating the discovery and optimization of novel, potent ADC payloads, particularly those targeting DNA Topoisomerase I, potentially leading to more effective and targeted cancer therapies.

### 2.5. Ablation Studies

To validate the theoretical principles underlying DumplingGNN’s architecture and elucidate the contribution of each component, we conducted a series of rigorous ablation studies. These experiments systematically removed or modified key components of our model, allowing us to quantify their individual and combined impacts on prediction performance. This approach not only demonstrates the empirical effectiveness of our hybrid architecture but also provides insights into how each component aligns with our theoretical motivations. Table 4 presents the results of these experiments on the ADC payload dataset.

For clear analysis and interpretation, we organized the model variants into the following three categories: Complete Architecture (the full DumplingGNN model), Input Representation Variants (examining the impact of using only SMILES data without 3D information), and Architectural Component Variants (systematically removing key layers such as GraphSAGE, GAT, or MPNN). This structured categorization allows us to assess both the contribution of different types of molecular information and the synergistic effects of various architectural components.

Our ablation studies reveal several critical insights that validate the theoretical principles guiding DumplingGNN’s architecture.

#### 2.5.1. Significance of 3D Structural Information

The most striking observation is the substantial performance degradation when the model is restricted to SMILES input (SMILES-Only variant). This variant, which excludes 3D conformational data, shows a dramatic decrease in accuracy (from 0.915 to 0.734) and MCC (from 0.829 to 0.461). The AUC-ROC also drops significantly from 0.955 to 0.782. These results empirically validate our theoretical motivation for incorporating 3D structural information, as discussed in Section 4.1. From a biological perspective, the 3D conformation of a molecule directly influences its binding affinity and specificity by determining how well the molecule can interact with the active site of its target protein. Such proper spatial arrangement is essential for the effective inhibition of DNA Topoisomerase I. By incorporating 3D data, DumplingGNN is better able to mimic the real-life interactions between inhibitors and their target, leading to more biologically relevant predictions.

The importance of 3D structural information is further underscored by our comparative analysis across different datasets, while the model achieved impressive results on datasets using only SMILES input (e.g., ToxCast with 78.2% ROC-AUC), the performance gains from incorporating 3D information on both the ADC payload dataset and BBBP dataset demonstrate the critical role of spatial arrangement in molecular property prediction. This observation aligns with the established principles in medicinal chemistry, where the three-dimensional conformation of molecules plays a crucial role in determining their biological activity and pharmacokinetic properties. The substantial performance improvement observed when incorporating 3D information suggests that our model’s ability to capture spatial relationships between atoms and functional groups significantly enhances its predictive capabilities, particularly for tasks that depend heavily on molecular conformation, such as target binding and membrane permeability.

#### 2.5.2. Hierarchical Architecture Analysis

The removal of individual GNN components (GraphSAGE, GAT, or MPNN) each led to performance deterioration, validating our theoretical framework of a multi-scale approach to molecular representation, as follows:

##### MPNN Layer

The No MPNN variant showed a substantial performance degradation, with accuracy falling to 0.812 and MCC to 0.619. This empirically confirms the importance of MPNN’s message-passing mechanism in effectively propagating and aggregating atomic and bond information across the molecular graph, as theorized in Section 4.3.1.

##### GAT Layers

Removing the Graph Attention layers (No GAT variant) led to a similar level of performance decline, with an accuracy at 0.803 and MCC at 0.601. This result validates our theoretical motivation for incorporating attention mechanisms, as outlined in Section 4.3.2. The substantial performance drop highlights the significance of GAT in weighing the importance of different molecular substructures for activity prediction.

##### GraphSAGE Layer

The exclusion of GraphSAGE layers (No GraphSAGE variant) resulted in a moderate performance drop, with accuracy decreasing from 0.915 to 0.870 and MCC from 0.829 to 0.737. This finding supports our theoretical framework presented in Section 4.3.3, where GraphSAGE was hypothesized to capture global molecular properties. The moderate impact suggests that while GraphSAGE plays an important role in aggregating information across different scales, some of its functionality may be partially compensated by the remaining architectures.

### 2.6. Model Interpretability and Biological Insights

To validate the biological relevance of DumplingGNN’s predictions, we conducted comprehensive analyses of the model’s attention mechanisms and their relationship to known pharmacophoric features of DNA Topoisomerase I inhibitors.

#### 2.6.1. Attention Mechanism Analysis

The analysis of attention patterns across layers reveals a progressive refinement of molecular feature recognition [47,48], as illustrated in Figure 1A. Our quantitative assessment demonstrates a clear evolution in the attention mechanism, as follows:Layer 0: Shows broad feature detection with 6.61 effective attention heads and high diversity (σ=0.282), reflecting the initial exploration of the feature space.Layer 1: Demonstrates increased focus with 7.93 effective heads and stabilizing attention patterns (σ=0.412), indicating convergence on the relevant chemical patterns.Layer 2: Achieves refined feature selection with 7.97 effective heads and consistent attention distribution (σ=0.429), suggesting mature feature recognition.

The attention entropy decreases across layers (4.846 → 4.342 → 4.262), while the standard deviation of attention scores increases slightly and then stabilizes (0.283 → 0.412 → 0.430), as visualized in Figure 1A(a). This pattern indicates a transition from broad feature exploration to focused pharmacophore recognition [49]. The increasing number of effective heads (Figure 1A(b)) coupled with the evolving head importance distribution (Figure 1A(c)) demonstrates how the model progressively refines its attention mechanisms.

Notably, the head importance distribution becomes more uniform as we move deeper into the network—from a highly variable distribution in layer 0 (with values ranging from 0.079 to 0.244, showing some heads strongly dominating others) to a much more balanced contribution in layer 2 (with values ranging from 0.118 to 0.130). This evolution reflects the model’s transition from relying on a few specialized attention heads to utilizing all heads more equally, creating a robust ensemble effect for feature recognition.

#### 2.6.2. Pharmacophore Recognition and Biological Validation

The model successfully identified key structural features crucial for DNA Topoisomerase I inhibition [27,50], as displayed in Figure 1B. Three critical pharmacophoric elements stood out in our analysis as follows:**Ester group (COC(=O))**: As highlighted in Figure 1B(a), this moiety showed consistently high attention scores (0.316±0.003) with a remarkably low standard deviation. This group serves as a crucial linking element that stabilizes the positioning of Topoisomerase I inhibitors in the binding site, aligning with binding modes observed in the crystallographic studies of clinically approved DNA Topoisomerase I inhibitors.**Hydroxyl group (C[OH])**: Shown in Figure 1B(b), this group received equivalent attention scores for the ester group (0.316±0.003) and functions primarily as a hydrogen bond donor/acceptor, facilitating critical interactions with specific amino acid residues in the binding pocket of DNA Topoisomerase I.**Five-membered ring system**: Depicted in Figure 1B(c), this structural element was identified across all 177 analyzed molecules with a significant attention score (0.222±0.028). Such a ring structure provides a rigid core scaffold that is often associated with planar aromatic systems for the enzyme inhibiting mechanism observed in DNA Topoisomerases. It provides the core scaffold essential for DNA intercalation and is a defining feature of camptothecin derivatives and other Topoisomerase I inhibitors currently used in clinical practice.

These findings match the established structure–activity relationships for Topoisomerase I inhibitors, with the model’s attention patterns highlighting the following three key pharmacophoric requirements: hydrogen bond donors/acceptors for target recognition, planar aromatic systems for DNA intercalation, and specific linker groups for optimal positioning within the binding site.

The biological relevance of these identified features is supported by extensive evidence [29,51], including crystallographic data, medicinal chemistry principles, and clinical drug design strategies. This comprehensive analysis demonstrates that DumplingGNN not only achieves high predictive accuracy but also provides biologically meaningful insights that can guide rational drug design for ADC payload optimization.

## 3. Discussion

### 3.1. Clinical and Drug Discovery Implications

DumplingGNN’s exceptional performance across diverse molecular property prediction tasks, particularly in toxicity assessment and ADC payload activity prediction, has significant implications for drug discovery and clinical development.

The model’s strong performance on the ToxCast dataset (78.2% ROC-AUC across 617 tasks) demonstrates its potential for comprehensive toxicity screening. This capability is crucial for early safety assessment in drug development, potentially reducing attrition rates in clinical trials due to unforeseen toxicity issues. Moreover, DumplingGNN’s ability to predict blood–brain barrier penetration (96.4% ROC-AUC on BBBP) could significantly impact the development of CNS-targeted therapies and the assessment of CNS side effects for non-CNS drugs.

Beyond predictive performance, DumplingGNN’s interpretability framework provides valuable insights for clinical applications [47,48]. The identification of key structural features, such as the ester group (CC(=O)O) and five-membered ring systems, aligns with established medicinal chemistry knowledge and offers practical guidance for cytotoxic drug design. The model’s attention analysis reveals the following:Structure–activity patterns: The consistent attention scores for key pharmacophoric elements (0.316±0.003 for essential groups) provide quantitative guidance for structural optimization [24,52]. At the local scale, individual atom interactions and bond connectivity determine immediate activity, similar to how an enzyme recognizes functional groups. At a global scale, the overall molecular properties affect cellular uptake and distribution which aligns our molecular-level understanding with the phenotypes. The hierarchical approach adopted here not only improves prediction accuracy but also resonates with the way medicinal chemists think about structure–activity relationships.Safety-related features: The identification of potentially toxic substructures helps in the early risk assessment and modification of problematic molecular features.Design guidelines: The hierarchical attention patterns across layers (diversity from 0.057 to 0.007) suggest a systematic approach to molecular optimization. The integration of interpretability (via attention mechanism analysis) provides quantitative guidance on which molecular features are most critical for activity. This is valuable for the future drug design process, as it translates the abstract numbers of a machine learning model into tangible design principles, such as optimizing the positioning of ester or hydroxyl groups to enhance binding affinity and reduce toxicity. With the correlation of the observed attention patterns with known pharmacophoric requirements, our model validates its predictions through established biological principles, thus strengthening the case for its practical use in early drug discovery.

In the context of ADC development, DumplingGNN’s high accuracy in predicting DNA Topoisomerase I inhibitor activity (91.48% accuracy and 97.54% specificity) could streamline the design and optimization of these complex therapeutics. The model’s ability to effectively distinguish between active and inactive compounds could reduce the time and resources invested in suboptimal candidates, a critical factor in ADC payload selection.

The model’s versatility, demonstrated by its consistent performance across diverse datasets, extends its applicability to a wide range of drug discovery tasks. Its high AUC-ROC and AUC-PR scores in large-scale virtual screening scenarios indicate excellent compound ranking ability, potentially saving considerable time and resources in drug discovery pipelines.

Furthermore, DumplingGNN’s incorporation of 3D structural information represents a significant advancement in capturing the spatial aspects of molecular interactions. This feature is particularly relevant for predicting binding affinities and understanding molecule–target interactions, offering a more comprehensive approach to in silico drug screening.

As a research prototype, DumplingGNN is being considered for integration into Omni Medical, an AI-driven drug discovery platform currently under development. This potential integration represents an initial step toward bridging the gap between computational predictions and experimental validation. However, it is important to note that the model’s current implementation does not include formal uncertainty quantification, which would be essential for clinical or regulatory applications. Future work will focus on incorporating robust uncertainty estimation methods and conducting extensive validation studies before considering any clinical deployment.

### 3.2. Limitations and Future Directions

Notwithstanding DumplingGNN’s robust performance, several limitations and avenues for future enhancement exist. A pivotal challenge is the inconsistent deployment of 3D structural data across datasets, while BBBP and our ADC payload datasets embraced comprehensive 3D conformational information, other datasets were constrained to SMILES representations. Future endeavors should prioritize the expansion of 3D data usage across all datasets, which could potentially augment the model’s performance on tasks where it currently exhibits room for improvement.

Another significant limitation of our current model is its focus on payload prediction without explicitly considering the influence of linker chemistry on payload conformation and activity. The linker–payload interface can meaningfully alter the 3D conformation of the payload, potentially affecting its downstream biological behavior. This is particularly evident in the documented success of specific linker–payload combinations, such as Exo-linkers with Topoisomerase I inhibitors in preclinical studies [53]. While our model achieves high accuracy in predicting payload activity in isolation, the complex interactions between linker and payload components represent an important area for future development.

The interpretability framework, while providing valuable insights, also faces certain limitations as follows [21]:Attention pattern complexity: The evolution of attention scores through multiple layers (entropy from 0.248 to 0.260) can be challenging to interpret comprehensively.Validation challenges: While the model identifies known pharmacophoric elements, validating novel structural insights requires extensive experimental confirmation.Scale limitations: The current framework’s effectiveness in analyzing very large molecules or complex protein–ligand interactions needs further investigation.

Another avenue for advancement is the integration of domain-specific knowledge. The incorporation of information regarding protein targets, cellular pathways, or known structure–activity relationships could serve to enhance the model’s predictive power and interpretability. This approach could prove particularly beneficial in the context of ADC payload prediction, where a comprehensive understanding of the interaction between the payload and the target protein is of paramount importance.

Further opportunities for improvement may be identified through architectural optimization. The exploration of additional GNN variants or the development of novel architectures specifically tailored for molecular property prediction could yield further enhancements. The investigation of more advanced attention mechanisms or the incorporation of recent advancements in transformer-based models for graph-structured data represent promising directions for future research.

Given the model’s strength in multi-task learning scenarios, future research could explore its application to an even broader range of molecular property prediction tasks. This expansion could include predicting ADMET properties, protein–ligand interactions, or even extending to related fields such as materials science. The model’s ability to handle complex, multi-task learning problems makes it well-suited for these challenging applications.

Looking ahead, we envision a comprehensive framework for ADC design that integrates different know-how requirements for each component. This framework will extend beyond the current focus on payload prediction to include the following: (1) For the antibody component, advanced target recognition strategies including masking technologies and internalization efficiency. (2) For the linker component, the optimization of cleavage efficiency and stability, with specific attention to the biological environment and cleavage mechanisms. (3) For the payload component, enhanced focus on efflux mechanisms and bystander killing effects. This multi-dimensional approach will be integrated in a MOE (Multi-Objective Evolution) style framework, allowing for the simultaneous optimization of different aspects of ADC design while maintaining the synergistic interaction between components. Such an integrated approach will better serve the needs of translational medicine by providing more practical and applicable solutions for ADC development.

In conclusion, while DumplingGNN represents a significant advancement in molecular property prediction, particularly for ADC payloads, there remain promising avenues for the further enhancement and expansion of its capabilities. The model’s robust performance, even with limited 3D structural information, suggests that future improvements in this area could yield even more impressive results, further advancing the field of computational drug discovery and development.

## 4. Materials and Methods

This section details the technical aspects of our proposed DumplingGNN model, including the construction of a novel ADC payload dataset, the method for molecular graph construction, the design of the network architecture, and the training and evaluation strategies.

### 4.1. Data: A Novel ADC Payload Dataset

Our dataset primarily focuses on Topoisomerase I inhibitors, which represent one of the most clinically significant classes of ADC payloads. This focus is scientifically justified by their established clinical relevance, as evidenced by their use in multiple FDA-approved ADCs. The well-characterized mechanisms of action involving DNA damage make these inhibitors particularly suitable for computational modeling, while their established structure–activity relationships provide a solid foundation for model validation. This specialized focus on a therapeutically important payload class allows our model to develop deep expertise as opposed to a superficial coverage across multiple mechanisms, which is crucial for building a robust and reliable prediction system. Molecules with a DNA Topoisomerase I inhibitory capability were initially collected from the CHEMBL database (accessed on 5 January 2024) (https://www.ebi.ac.uk/chembl/) using the identifier CHEMBL1781, yielding a preliminary set of 2292 compounds. The data profile was thoroughly processed to reduce redundancy and maintain high-quality presentation, resulting in 190 unique molecular structures. To develop a well-performing classification model, a cutoff value of 100 µM was established, with inhibitory IC50 values below this threshold labeled as positive and those above that labeled as negative.

The curated set of 190 unique molecules possessing potent inhibitory effects, specifically against DNA Topoisomerase I, along with their topological information presented in the form of SMILES strings, were prepared for reliable 3D conformation generation. Initial conformer generation was performed using OpenBabel [54] and minimized under the MMFF94 force field via RDKit [55]. Conformer redundancy was eliminated using an RMSD cutoff of 1.5 Å. To determine the coordinate space of the potential Topoisomerase I inhibitors, NLDock [56] was employed to generate binding conformations in the binding pocket of crystallized Topoisomerase I, localized with co-crsytalized camptothecin (PDB entry ID: 1t8i [57]). The top three docking poses for each conformer, ranked by estimated binding energy, were saved for model construction, resulting in a total of 885 distinct 3D conformations.

To further enrich our dataset, we extracted an additional 615 molecular structures from recent ADC payload-related patents, significantly expanding the chemical diversity and real-world relevance of our collection. This multifaceted approach yields a first-of-its-kind ADC payload dataset that synergistically integrates experimental data, high-quality computational predictions, and state-of-the-art industrial insights.

The dataset’s innovation lies in its comprehensive representation of molecular structures, spanning from 2D topological to 3D conformations curated based on active interaction patterns revealed by the docking study, as well as its integration of public database entries with proprietary patent-derived structures. By incorporating molecules at various stages of the drug discovery pipeline, from initial screens to patented compounds, this dataset offers unprecedented relevance to real-world applications in ADC payload development. It not only provides a robust foundation for the development and evaluation of our DumplingGNN model but also represents a valuable resource for the broader ADC research community, potentially accelerating the discovery and optimization of novel ADC payloads.

### 4.2. Molecular Graph Construction

In order to transform ADC payload small molecules into a format suitable for graph neural network processing, we designed an enhanced molecular graph construction algorithm [13]. Unlike traditional methods that only extract atom type information, our algorithm also incorporates chemical environment information to more comprehensively characterize the molecular structure [17].

Specifically, for a given molecule (mol), we extract the following atomic features:–**Atomic number**: Indicates the type of element.–**Degree**: The number of bonds attached to the atom–**Number of hydrogen atoms**: Number of hydrogen atoms attached to the atom.–**Implicit valence**: The atomic valence level at which all chemical bonds are considered.–**Aromaticity judgement**: Whether the atom belongs to an aromatic ring or not.–**Atomic coordinates**: The position of the atom in three-dimensional space.

These features reflect the chemical properties of atoms from different perspectives and help the graph neural network to understand the molecular structure more accurately [10]. In particular, the introduction of atomic coordinate information enables the model to capture the spatial conformation of the molecule, which is crucial for the conformational relationship of ADC payload [23].

While extracting the atomic node features, we also construct the edge index information, i.e., the existence of an undirected edge between each pair of connected atoms [16]. Eventually, the atom features and edge indexes are converted into PyTorch (version 2.0.0) tensor and encapsulated into PyTorch Geometric’s Data object along with the molecule’s activity labels [13]. Compared to traditional molecular graph construction methods that only consider atom types and chemical bonds, our algorithm incorporates richer chemical information, enabling downstream graph neural networks to learn more effective molecular representations from the data, thus improving prediction performance [17,22].

### 4.3. Network Architecture Design

Based on the molecular graph data constructed above, we designed an innovative graph neural network model, DumplingGNN, which adopts a hybrid architecture to leverage the strengths of different GNN modules [13,16]. The workflow of DumplingGNN’s architecture, illustrated in Figure 2, integrates MPNN, GAT, and GraphSAGE layers to capture multi-scale molecular features. This design is motivated by the complex nature of ADC payload activity prediction, which requires capturing both local chemical interactions and global molecular properties.

The structure of the **DumplingGNN** model consists of an MPNN-GAT*3-SAGE sequence, integrating multiple powerful GNN components to process the molecular data effectively [17,22,58]. This sequential design allows the model to progressively refine and aggregate molecular information at different scales, mimicking the hierarchical nature of chemical interactions in biological systems.

#### 4.3.1. Message Passing Neural Network (MPNN) Layer

The first layer of DumplingGNN is a Message Passing Neural Network (MPNN) [17], which aggregates information from neighboring nodes through a message-passing mechanism, as illustrated in Figure 3.(1)$$h_{i}^{\left(k+1\right)}=\sigma \left(W^{\left(k\right)}h_{i}^{\left(k\right)}+\sum_{j\in N(i)}M^{\left(k\right)}\left(h_{i}^{\left(k\right)},h_{j}^{\left(k\right)}\right)\right)$$

The MPNN layer serves as the foundation of our model, capturing local chemical interactions within the molecular graph. This aligns with the principle of chemical locality, where an atom’s properties are primarily influenced by its immediate chemical environment. In the context of ADC payload activity prediction, this layer can be interpreted as modeling the local reactivity and functional group interactions that contribute to the payload’s overall activity.

#### 4.3.2. Graph Attention Network (GAT) Layers

Following the MPNN layer, we apply three Graph Attention Network (GAT) layers to adaptively assign weights to different neighbors using an attention mechanism, as illustrated in Figure 4 [22,59]. This approach captures key chemical structure information in the molecular graph as follows:(2)$$e_{ij}^{\left(k\right)}=\mathrm{LeakyReLU}\left(a^{T}\left[W^{\left(k\right)}h_{i}^{\left(k\right)}\|W^{\left(k\right)}h_{j}^{\left(k\right)}\right]\right)$$(3)$$\alpha_{ij}^{\left(k\right)}=\frac{\exp(e_{ij}^{\left(k\right)})}{\sum_{k\in N(i)}\exp\left(e_{ik}^{\left(k\right)}\right)}$$(4)$$h_{i}^{\left(k+1\right)}=\sigma \left(\sum_{j\in N(i)}\alpha_{ij}^{\left(k\right)}W^{\left(k\right)}h_{j}^{\left(k\right)}\right)$$

The Graph Attention Network (GAT) layers in DumplingGNN selectively emphasize important atomic interactions, aligning with the pharmacophore concept in medicinal chemistry [48]. This approach allows the model to identify critical substructures influencing ADC payload activity. The three sequential GAT layers enable the capture of complex, multi-scale molecular interactions [60], crucial for understanding how molecular conformation and functional group distribution affect ADC payload efficacy.

Each GAT layer employs an 8-head attention mechanism, where each attention head independently learns to focus on different aspects of molecular structure. The multi-head design provides several advantages, outlined as follows:Diverse feature capture: Each head can specialize in different chemical patterns.Robust feature extraction: Multiple perspectives reduce the risk of missing important features.Enhanced stability: Averaging across heads provides more stable attention scores.Improved interpretability: Different heads can reveal various aspects of molecular recognition.

The attention weights evolve across the three GAT layers, forming a hierarchical feature extraction process as follows:First layer: Focuses on atomic-level features (mean attention = 0.246, std = 0.283).Second layer: Recognizes functional groups and local patterns (mean attention = 0.284, std = 0.412).Third layer: Captures global structural features (mean attention = 0.316, std = 0.430).

By analyzing attention coefficients, we can visualize key atomic interactions and molecular fragments prioritized by the model. This transforms DumplingGNN into a transparent tool for drug discovery, providing medicinal chemists with valuable insights for lead optimization. The interpretability offered by GAT layers opens new research avenues, potentially guiding automated molecular design and focusing on refining critical substructures. Comparing attention patterns across different ADC payload classes may reveal common structural features underlying their mechanisms, extending insights to other drug discovery areas.

The GAT layers in DumplingGNN not only enhance predictive performance but also provide crucial interpretability. This feature is essential for building trust in the model’s predictions and deriving actionable insights for ADC payload optimization. Future studies could correlate attention signatures with known binding sites from crystallographic data, bridging predictive analytics and structural biology. Ultimately, this approach paves the way for more efficient and targeted drug discovery efforts, deepening our understanding of structure–activity relationships in ADC payloads.

#### 4.3.3. GraphSAGE Layer

The final layer of our architecture is a GraphSAGE layer, which effectively aggregates node information from different domains to learn multi-scale representations of molecules through sampling and aggregation, as illustrated in Figure 5 [58].(5)$$h_{i}^{\left(k+1\right)}=\sigma \left(W^{\left(k\right)}h_{i}^{\left(k\right)}+\sum_{j\in N(i)}\frac{1}{\left|N(i)\right|}W^{\left(k\right)}h_{j}^{\left(k\right)}\right)$$

The GraphSAGE layer serves to capture global molecular properties by aggregating information across different scales. In the context of ADC payload prediction, this can be interpreted as modeling the overall molecular properties that influence activity, such as lipophilicity, molecular weight, and topological features. These global properties are crucial for predicting how the payload will behave in a biological system, including its ability to penetrate cell membranes and interact with the target.

### 4.4. Synergistic Effects and Biological Interpretability

The sequential combination of MPNN, GAT, and GraphSAGE layers in DumplingGNN allows for a hierarchical and comprehensive analysis of the molecular structure and properties, as follows:The MPNN layer captures local chemical interactions, modeling the reactivity and functional group behavior of the ADC payload.The GAT layers identify key substructures and atomic relationships, mimicking the concept of pharmacophores in drug discovery.The GraphSAGE layer aggregates information to model global molecular properties, which are crucial for predicting the payload’s behavior in biological systems.

This multi-scale approach aligns well with the complex nature of ADC payload activity, which depends on both local chemical reactivity and global molecular properties. The model’s ability to capture these different levels of molecular information enables it to make more accurate and biologically relevant predictions.

To ensure biological interpretability, DumplingGNN implements a comprehensive analysis framework (Figure 6). The attention score collection process operates across all layers as follows:Layer-wise attention collection: Each layer’s attention weights are collected and normalized, providing insights into the model’s focus at different abstraction levels.Multi-scale feature analysis: Attention patterns reveal the hierarchical recognition of molecular features, from atomic properties to global structural characteristics.Temporal evolution tracking: The progression of attention scores through layers demonstrates how the model builds its understanding of molecular properties.

The biological interpretation process integrates multiple analytical approaches as follows:Attention score analysis: Identifies atoms and bonds crucial for activity prediction, highlighting potential pharmacophoric elements.SMARTS pattern matching [61]: Recognizes functional groups and their relative importance based on attention scores.Murcko scaffold analysis [62]: Evaluates the contribution of core molecular frameworks to predicted activity.BRICS decomposition [63]: Identifies key fragments and their hierarchical relationships in molecular recognition.

This interpretability framework provides valuable insights for medicinal chemistry as follows:Structure–activity relationships: Understanding which molecular features contribute most significantly to the activity.Design guidelines: Identifying preferred scaffolds and functional groups for optimization.Mechanism insights: Revealing potential binding modes and interaction patterns.Safety assessment: Highlighting structural features that might contribute to toxicity.

To obtain the final activity prediction for each molecule, we apply global average pooling followed by a linear layer. This approach, commonly used in graph neural networks for molecular property prediction [17,19], allows our hybrid architecture to effectively aggregate node-level information into a graph-level representation. Consequently, DumplingGNN captures both local and global structural information, enhancing its ability to predict ADC payload activity accurately.

The interpretability of DumplingGNN is further enhanced by the attention mechanisms in the GAT layers, which can highlight the atomic interactions most crucial for activity prediction. This feature not only improves prediction accuracy but also provides valuable insights for medicinal chemists in the design and optimization of ADC payloads.

## 5. Conclusions

This study introduces DumplingGNN, a novel hybrid Graph Neural Network model that significantly advances molecular property prediction, with a particular focus on Antibody–Drug Conjugate (ADC) payload cytotoxic activity. Our comprehensive evaluation across multiple datasets demonstrates the model’s exceptional performance and versatility. DumplingGNN’s innovative architecture, synergistically combining Message Passing Neural Networks, Graph Attention Networks, and GraphSAGE, captures complex molecular features and interactions with unprecedented accuracy.

Beyond predictive accuracy, DumplingGNN’s interpretability framework provides crucial insights into molecular recognition mechanisms. The model’s attention analysis reveals a progressive refinement of feature detection (entropy: 0.248 → 0.260) and successfully identifies key pharmacophoric elements, including the essential ester group (attention score: 0.316±0.003) and five-membered ring systems (attention score: 0.222±0.028). These findings align with established medicinal chemistry knowledge and provide quantitative guidance for molecular optimization.

This hybrid approach, integrating 3D structural information where available, achieves state-of-the-art performance across diverse tasks, notably attaining 91.48% accuracy on our specialized ADC payload dataset and setting new benchmarks on public datasets such as BBBP (96.4% ROC-AUC), ToxCast (78.2% ROC-AUC), and PCBA (88.87% ROC-AUC).

The biological validation of our model’s predictions, particularly in identifying known pharmacophoric features of Topoisomerase I inhibitors, demonstrates DumplingGNN’s ability to capture meaningful structure–activity relationships. The hierarchical attention patterns across layers (diversity from 0.057 to 0.007) provide insights into the model’s feature recognition process, offering a systematic approach to understanding molecular recognition in drug–target interactions.

A key innovation lies in DumplingGNN’s effective utilization of 3D structural information and its integration of attention mechanisms, providing the strong interpretability crucial for drug discovery and optimization. As a research prototype, DumplingGNN represents an important step toward bridging computational prediction and experimental validation. The model’s current capabilities, while promising, require further development and validation before considering clinical or regulatory applications.

Looking ahead, DumplingGNN opens several exciting avenues for future research, while our current work focuses primarily on Topoisomerase I inhibitors due to their clinical significance and data availability, we envision expanding the model’s capabilities to encompass other payload mechanisms, including tubulin inhibitors, DNA cross-linkers, and emerging payload classes. This expansion will be complemented by enhanced 3D structural information utilization and the integration of additional domain-specific knowledge. A particularly important direction involves incorporating explicit uncertainty quantification methods such as Bayesian approaches, which will significantly enhance the model’s utility in clinical and regulatory contexts.

To address the limitations related to linker–payload interactions, we plan to develop methods for modeling the payload–linker interface and its impact on payload conformation and activity. This will involve creating a unified graph representation that captures the interactions between payload and linker components, as well as extending the model to generate and analyze conformers of complete ADC structures. The architecture will be enhanced to incorporate linker-specific features and their influence on payload activity, enabling more comprehensive predictions for ADC design and optimization.

A key aspect of this development will be the incorporation of different linker modalities and their specific optimization requirements. For cleavable linkers, we will focus on modeling cleavage efficiency and plasma stability, taking into account the biological environment and cleavage mechanisms. For non-cleavable linkers, we will emphasize circulatory stability and intracellular processing efficiency. This comprehensive approach will allow us to better understand and predict the behavior of different ADC designs, ultimately leading to more effective therapeutic strategies.

These developments will build upon the solid foundation established by the current work, leveraging the demonstrated strengths of our hybrid architecture while exploring novel applications in related fields.

In conclusion, DumplingGNN not only advances ADC payload activity prediction but also offers a versatile approach to molecular property prediction in general. As AI-driven drug discovery evolves, models like DumplingGNN are poised to play a critical role in shaping pharmaceutical research and development. By providing computational insights to guide experimental design, DumplingGNN contributes to transforming drug discovery into a more efficient and cost-effective process, while acknowledging the need for further validation and development before clinical deployment.

## Figures and Tables

**Figure 1 ijms-26-04859-f001:**
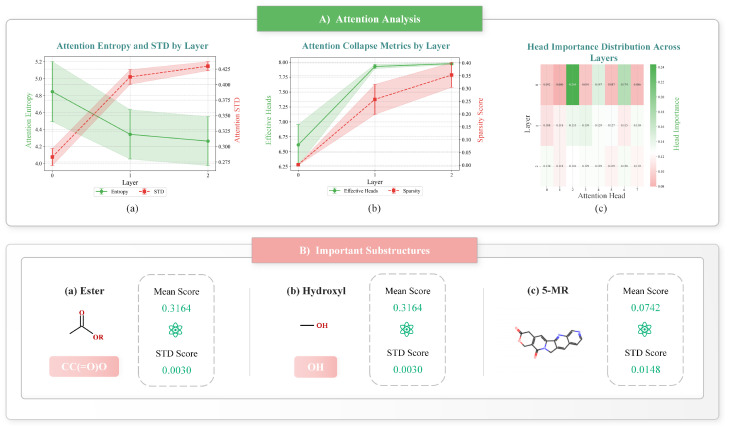
Interpretability analysis results from DumplingGNN. (**A**) Attention analysis showing entropy and standard deviation by layer (**a**), attention collapse metrics by layer (**b**), and head importance distribution across layers (**c**). (**B**) Important substructures identified by the model with their mean and standard deviation scores, including ester group (**a**), hydroxyl group (**b**), and five-membered ring ((**c**) 5-MR).

**Figure 2 ijms-26-04859-f002:**
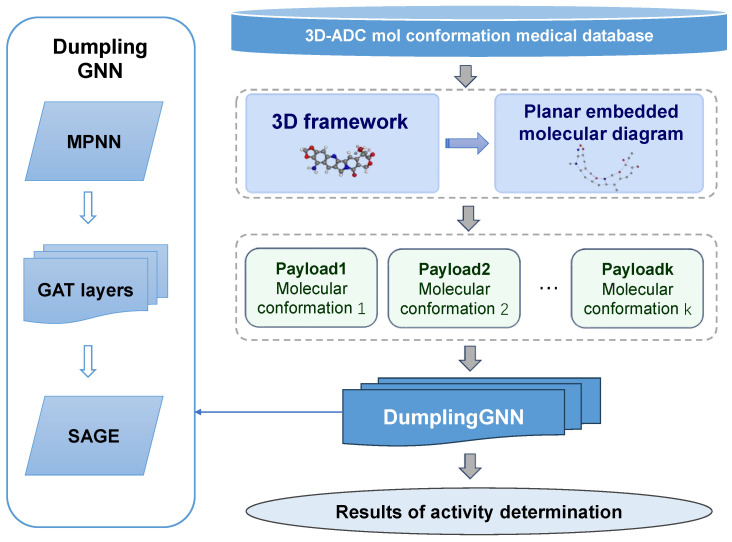
Workflow of DumplingGNN’s architecture for ADC payload activity prediction. The hybrid architecture integrates MPNN, GAT, and GraphSAGE layers to capture multi-scale molecular features.

**Figure 3 ijms-26-04859-f003:**
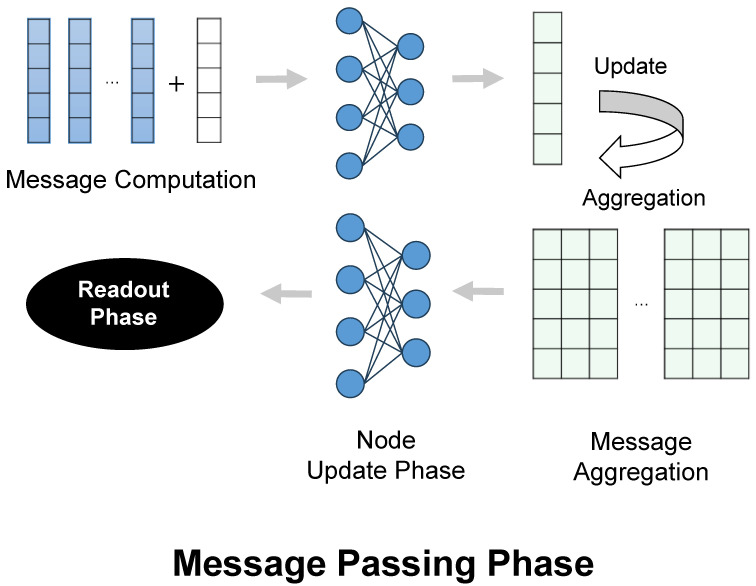
Process of message computation and node update in MPNN. This mechanism allows for the modeling of local chemical interactions.

**Figure 4 ijms-26-04859-f004:**
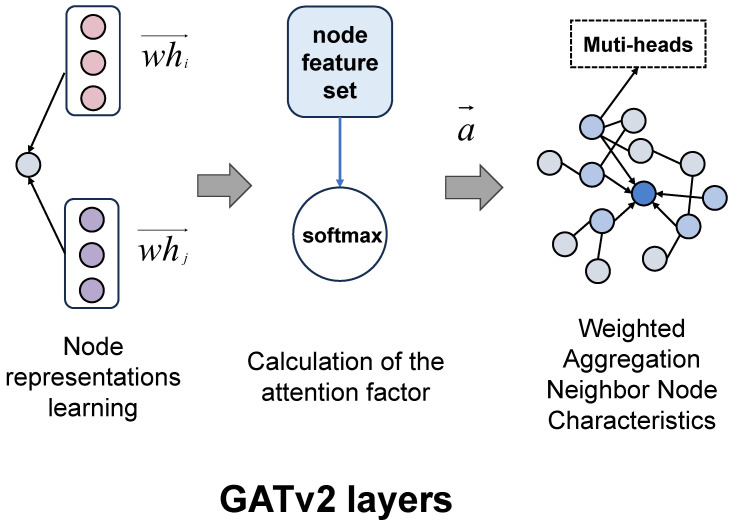
Graph Attention Network (GAT) in DumplingGNN. The attention mechanism allows the model to focus on the most relevant atomic interactions for activity prediction, enhancing both performance and interpretability.

**Figure 5 ijms-26-04859-f005:**
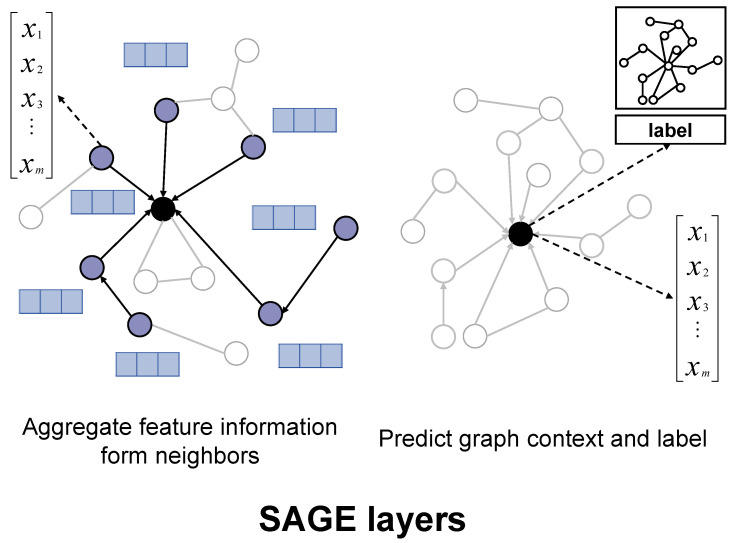
GraphSAGE layer in DumplingGNN. This layer aggregates information across different scales, capturing global molecular properties.

**Figure 6 ijms-26-04859-f006:**
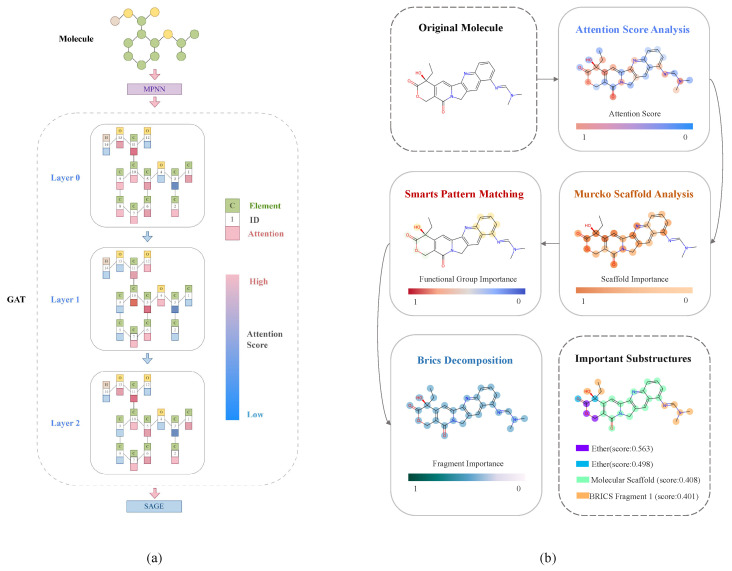
Interpretability analysis workflow in DumplingGNN. (**a**) Layer-wise attention score collection and visualization process. (**b**) Biological interpretation through attention score analysis, SMARTS pattern matching, Murcko scaffold analysis, and BRICS decomposition.

**Table 1 ijms-26-04859-t001:** Overview of datasets used in the experiments.

Dataset	Molecules	Tasks	Description
BBBP	2039	1	Blood–Brain Barrier Penetration
BACE	1513	1	Inhibition of Beta-Secretase 1
ClinTox	1478	2	Clinical Trial Toxicity
Tox21	7831	12	Toxicology in the 21st Century
ToxCast	8575	617	EPA Toxicity Forecaster
SIDER	1427	27	Drug Side Effect Resource
HIV	41,127	1	HIV Replication Inhibition
PCBA	437,929	128	PubChem Bioassay Data

**Table 2 ijms-26-04859-t002:** Performance comparison (ROC-AUC %, higher is better) on MoleculeNet datasets.

Model Category	Model	BBBP	BACE	ClinTox	Tox21	ToxCast	SIDER	HIV	PCBA
**Proposed Model**	DumplingGNN	**96.4 (0.7)**	88.2 (0.5)	95.9 (2.0)	82.3 (0.4)	**78.2 (0.1)**	74.0 (0.6)	79.4 (0.2)	**88.87 (0.2)**
**Single-Architecture GNNs**	D-MPNN [34]	71.0 (0.3)	80.9 (0.6)	90.6 (0.6)	75.9 (0.7)	65.5 (0.3)	57.0 (0.7)	77.1 (0.5)	86.2 (0.1)
	Attentive FP [23]	85.5	78.4 (0.022)	84.7 (0.3)	76.1 (0.5)	63.7 (0.2)	60.6 (3.2)	75.7 (1.4)	80.1 (1.4)
	GraphConv + dummy super node [40]	–	–	–	**85.4**	76.8	–	85.1	86.7
**Traditional ML**	N-Gram RF [35]	69.7 (0.6)	77.9 (1.5)	77.5 (4.0)	74.3 (0.4)	–	66.8 (0.7)	77.2 (0.1)	–
	N-Gram xGB [35]	69.1 (0.8)	79.1 (1.3)	87.5 (2.7)	75.8 (0.9)	–	65.5 (0.7)	78.7 (0.4)	–
**Pre-trained Models**	PretrainGNN [41]	68.7 (1.3)	84.5 (0.7)	72.6 (1.5)	78.1 (0.6)	65.7 (0.6)	62.7 (0.8)	79.9 (0.7)	86.0 (0.1)
	GROVER_large_ [36]	69.5 (0.1)	81.0 (1.4)	76.2 (3.7)	73.5 (0.1)	65.3 (0.5)	65.4 (0.1)	68.2 (1.1)	83.0 (0.4)
	GraphMVP [37]	72.4 (1.6)	81.2 (0.9)	79.1 (2.8)	75.9 (0.5)	63.1 (0.4)	63.9 (1.2)	77.0 (1.2)	–
	MolCLR [42]	72.2 (2.1)	82.4 (0.9)	91.2 (3.5)	75.0 (0.2)	–	58.9 (1.4)	78.1 (0.5)	–
	GEM [43]	72.4 (0.4)	85.6 (1.1)	90.1 (1.3)	78.1 (0.1)	69.2 (0.4)	67.2 (0.4)	80.6 (0.9)	86.6 (0.1)
	Uni-Mol [44]	72.9 (0.6)	85.7 (0.2)	91.9 (1.8)	79.6 (0.5)	69.6 (0.1)	65.9 (1.3)	80.8 (0.3)	88.5 (0.1)
**Advanced Approaches**	MolXPT [38]	80.5 (0.5)	**88.4**	95.3 (0.2)	77.1	–	71.7	78.1	–
	ChemBFN [39]	95.74	73.56	**99.18**	–	–	–	79.37	–
**Previous SOTA**	(Best reported results)	95.74 [39]	**88.4** [38]	**99.18** [39]	**89.9** [45]	77.7 [44]	**91.1** [45]	**80.8** [35]	88.5 [44]

**Table 3 ijms-26-04859-t003:** Performance comparison of models on the ADC payload test set.

Architecture Type	Model	Accuracy	Sensitivity	Specificity	MCC	AUC-ROC	F1 Score	Balanced Accuracy	AUC-PR
**Hybrid Architecture**	DumplingGNN	**0.9148**	**0.9508**	**0.9754**	**0.8287**	**0.9547**	**0.9243**	**0.9111**	**0.9531**
**Single-Architecture GNNs**	FiveLayerMPNN	0.8655	0.9262	0.7921	0.7301	0.9281	0.8828	0.8592	0.9342
	FiveLayerGAT	0.8565	0.9180	0.7822	0.7117	0.8741	0.8750	0.8501	0.8653
	FiveLayerSAGE	0.7982	0.8525	0.7327	0.5917	0.8509	0.8221	0.7926	0.8668
	FiveLayerGCN	0.7623	0.8525	0.6535	0.5197	0.8301	0.7969	0.7530	0.8589

**Table 4 ijms-26-04859-t004:** Ablation study results on the ADC payload dataset.

Variant Category	Model Variant	Accuracy	Sensitivity	Specificity	MCC	AUC-ROC	F1 Score	Balanced Accuracy	AUC-PR
**Complete Architecture**	Full DumplingGNN	**0.915**	**0.951**	**0.975**	**0.829**	**0.955**	**0.924**	**0.911**	**0.953**
**Input Representation**	SMILES-Only	0.734	0.768	0.949	0.461	0.782	0.775	0.721	0.786
**Component Ablation**	No GraphSAGE	0.870	0.902	0.832	0.737	0.940	0.884	0.867	0.939
	No GAT	0.803	0.869	0.723	0.601	0.861	0.828	0.796	0.882
	No MPNN	0.812	0.869	0.743	0.619	0.878	0.835	0.806	0.840

## Data Availability

The data presented in this study are available upon request from the corresponding author. The core code and training data for DumplingGNN have been made available in a public GitHub repository at https://github.com/Iayce/DumplingGnn (accessed on 9 April 2025).

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
