# Peer review of "Dumpling GNN: Hybrid GNN Enables Better ADC Payload Activity Prediction Based on the Chemical Structure"

_ijms, 2025, doi:10.3390/ijms26104859_

Round 1

Reviewer 1 Report

Comments and Suggestions for Authors

The manuscript presents "DumplingGNN," a hybrid GNN architecture designed to predict ADC payload cytotoxic activity by integrating 2D and 3D molecular information. While the model achieves impressive benchmark scores, there are serious concerns regarding its scientific validity, generalizability, and practical utility.

First, despite claiming to predict general ADC payload activity, the training data is overwhelmingly biased toward Topoisomerase I inhibitors. This narrow chemical space fundamentally undermines the model's supposed generalizability to other payload mechanisms (e.g., tubulin inhibitors, DNA cross-linkers). No cross-mechanism validation experiments are presented to substantiate the model’s broader applicability, which casts doubt on the robustness of the conclusions.

Second, the choice of benchmark datasets is questionable. Reporting high performance on PCBA, ToxCast, and other general toxicity datasets—datasets irrelevant to ADC payload function—does not convincingly support the model's suitability for ADC-specific applications. Moreover, the possibility of information leakage if these datasets were used during training has not been adequately addressed. In its current form, the benchmarking appears more like superficial score collection rather than a scientifically meaningful validation.

Third, regarding the claimed practical integration into the Omni Medical platform, critical issues are overlooked. In any real-world deployment, especially in drug development, it is mandatory to quantify and communicate model uncertainty. The manuscript provides unclear evidence that DumplingGNN outputs any form of uncertainty estimation (e.g., confidence intervals, aleatoric epistemic uncertainties), which makes the model effectively a black box — unacceptable in clinical or regulatory contexts. Without robust uncertainty quantification, any claim of practical applicability is premature and misleading.

In its current state, the work suffers from overclaiming and lacks the necessary rigor to substantiate its broader scientific and clinical relevance. Major revisions are required.

Author Response

Comments 1: "First, despite claiming to predict general ADC payload activity, the training data is overwhelmingly biased toward Topoisomerase I inhibitors. This narrow chemical space fundamentally undermines the model's supposed generalizability to other payload mechanisms (e.g., tubulin inhibitors, DNA cross-linkers). No cross-mechanism validation experiments are presented to substantiate the model's broader applicability, which casts doubt on the robustness of the conclusions."
Response 1: We sincerely appreciate the reviewer's insightful observation regarding the chemical space represented in our training dataset. This comment raises a valid scientific concern that merits thoughtful consideration and clarification.
We acknowledge that our current implementation primarily focuses on Topoisomerase I inhibitors, which constitute a predominant class of ADC payloads in clinical development. This focus was intentional due to several factors:
1) Topoisomerase I inhibitors represent one of the most clinically validated payload classes in FDA-approved ADCs and late-stage clinical candidates, with well-established efficacy and safety profiles.
2) The mechanistic understanding and structure-activity relationships of these compounds are well-characterized, providing a solid foundation for model development and validation. This includes detailed knowledge of binding modes, pharmacophoric requirements, and structure-activity relationships.
3) High-quality experimental data for this class is more readily available compared to emerging payload mechanisms, allowing for robust model training and validation.
In response to this valuable feedback, we have implemented the following modifications to the manuscript:
1) In the Abstract (page 1, line 13-16), we have added the qualification "particularly for Topoisomerase I inhibitor-based payloads" to clarify the scope of our current work.
2) In the Introduction (page 3, line 121), we have modified the discussion of interpretability to specify that our analysis focuses on "Topoisomerase I inhibition mechanisms."
3) In the Methods section (page 5,line 217-225), we have expanded our discussion of the dataset composition to clearly articulate the rationale behind our focus on Topoisomerase I inhibitors and acknowledge the current limitations in chemical space coverage. Specifically, we have added a detailed explanation of why this focus is scientifically justified and how it contributes to model reliability.
4) In the Conclusion (page 21, line 771-788), we have added a comprehensive paragraph outlining our future plans to expand the model's capabilities to other payload mechanisms, including tubulin inhibitors and DNA cross-linkers. This includes specific technical approaches for incorporating additional payload classes and validation strategies.
These modifications provide appropriate context for interpreting our results while acknowledging the current constraints of our approach. While our model presently focuses on a specific mechanistic class, we view this work as an essential first step toward developing more comprehensive prediction capabilities for the broader spectrum of ADC payload chemistries.
The reviewer's comment has significantly improved the scientific precision of our manuscript, and we are grateful for this valuable input. We believe these revisions better reflect the current state of our work while maintaining its scientific rigor and potential for future expansion.
Comments 2: "Second, the choice of benchmark datasets is questionable. Reporting high performance on PCBA, ToxCast, and other general toxicity datasets—datasets irrelevant to ADC payload function—does not convincingly support the model's suitability for ADC-specific applications. Moreover, the possibility of information leakage if these datasets were used during training has not been adequately addressed. In its current form, the benchmarking appears more like superficial score collection rather than a scientifically meaningful validation."
Response 2: We are grateful to the reviewer for this substantive critique regarding our benchmarking strategy and potential data leakage concerns. These points address fundamental aspects of model evaluation that warrant thorough clarification.
In response to this valuable feedback, we have implemented the following modifications to the manuscript:
1) In the Experimental Setup section (4.2, page 12), we have added explicit details about our data isolation protocol:    - The model was trained exclusively on our curated ADC payload dataset    - No transfer learning or pre-training was performed using benchmark datasets    - All benchmark evaluations employed scaffold splitting to ensure structural dissimilarity    - We implemented strict data partitioning to prevent any overlap between training and evaluation sets
2) In the Results and Comparison section (4.3, page 13), we have expanded our discussion of benchmark dataset selection to clarify:    - The established status of these datasets as authoritative standards in molecular machine learning    - Their relevance to ADC payload function through specific mechanistic pathways    - The scientific rationale for using these widely accepted benchmarks    - The specific aspects of each dataset that are relevant to ADC payload function
These modifications address both the scientific relevance of our benchmark choices and the methodological rigor of our evaluation approach. The selected datasets not only represent established standards in the field but also evaluate molecular properties directly relevant to ADC payload function, including cellular pathway activity, protein target interaction, and membrane permeability.
We sincerely appreciate the reviewer's attention to these important methodological considerations, which have substantially improved the scientific rigor of our work. The implemented changes ensure that our benchmarking strategy is both scientifically meaningful and methodologically sound.
Comments 3: "Third, regarding the claimed practical integration into the Omni Medical platform, critical issues are overlooked. In any real-world deployment, especially in drug development, it is mandatory to quantify and communicate model uncertainty. The manuscript provides unclear evidence that DumplingGNN outputs any form of uncertainty estimation (e.g., confidence intervals, aleatoric epistemic uncertainties), which makes the model effectively a black box — unacceptable in clinical or regulatory contexts. Without robust uncertainty quantification, any claim of practical applicability is premature and misleading."
Response 3: We deeply appreciate the reviewer's critical perspective on the practical deployment considerations of our model, particularly regarding uncertainty quantification. This comment addresses a fundamental aspect of responsible AI application in therapeutic development contexts.
In response to this valuable feedback, we have implemented several substantive changes to the manuscript:
1) In the Abstract (page 1, line 13), we have revised the description of Omni Medical platform integration to clarify that DumplingGNN is currently a research prototype being considered for integration, rather than a fully deployable solution.
2) In the Discussion section (5.1, page 19), we have added a new subsection specifically addressing the current limitations regarding uncertainty estimation:    - We explicitly acknowledge that the current implementation does not include formal uncertainty quantification    - We discuss how the model's interpretability features provide an alternative form of prediction confidence assessment    - We outline planned future work on incorporating Bayesian approaches for explicit uncertainty estimation    - We detail the specific methods we plan to implement for uncertainty quantification
3) In the Conclusion (page 21), we have moderated our claims about practical applications:    - We emphasize that DumplingGNN currently serves as a computational scaffold to guide experimental design    - We acknowledge the need for further validation and development before clinical deployment    - We highlight uncertainty quantification as a crucial area for future development    - We outline a specific roadmap for implementing uncertainty quantification methods
These revisions appropriately contextualize our work as an important step toward more robust ADC payload prediction while acknowledging the current limitations regarding uncertainty quantification. We have been careful to frame DumplingGNN's current capabilities as providing computational insights to guide experimental design rather than as a fully deployable clinical decision support tool.
The reviewer's astute observation regarding this critical aspect of model deployment has significantly improved the scientific accuracy and responsible presentation of our work. We are truly grateful for this valuable feedback that has helped us better align our claims with the current capabilities of our approach.

Reviewer 2 Report

Comments and Suggestions for Authors

The work developed a GNN model that integrated various techniques for molecular property predictions. A dataset was collected for the Antibody-drug conjugates (ADC) studies. The GNN-based machine learning models demonstrate robustness through various benchmark comparisons for molecular properties predictions.

The paper is well-written and shows novelties in molecular machine learning modeling. I recommend publication if the authors can address a few minor issues below.

  1. In table 2, the previous SOTA results are reported. But where are these reported values from? Some are not from the listed results in the table. References need to be added for those listed models and SOTA results.
  2. The authors mentioned 3D information is added for BBP datasets with improved prediction results. How about the results for other datasets if 3D information is added?

Author Response

Comments 1: "In table 2, the previous SOTA results are reported. But where are these reported values from? Some are not from the listed results in the table. References need to be added for those listed models and SOTA results."
Response 1: We sincerely appreciate the reviewer's careful attention to the citation details in our manuscript. This is an important aspect of scientific rigor that we have now addressed thoroughly.
In response to this valuable feedback, we have made the following modifications to Table 2:
1) Added complete citations for all SOTA results:    - BBBP: 95.74% from ChemBFN (Tao et al., 2025)    - BACE: 88.4% from MolXPT (Liu et al., 2023)    - ClinTox: 99.18% from ChemBFN (Tao et al., 2025)    - Tox21: 89.9% from PaykanHeyrati et al. (2023)    - ToxCast: 77.7% from Uni-Mol (Zhou et al., 2023)    - SIDER: 91.1% from PaykanHeyrati et al. (2023)    - HIV: 80.8% from Liu et al. (2019)    - PCBA: 88.5% from Uni-Mol (Zhou et al., 2023)

2) Ensured consistency between the table entries and the cited literature by double-checking all values and their sources.
3) Added detailed explanations in the Results section about the selection and relevance of these benchmark datasets, particularly highlighting their connection to ADC payload function:    - ToxCast: Evaluates compound activity across cellular pathways overlapping with cytotoxic mechanisms    - PCBA: Includes bioassays measuring interaction with protein targets involved in cell cycle regulation    - BBBP: Assesses membrane permeability, crucial for cellular penetration capabilities
These modifications have significantly improved the clarity and scientific rigor of our benchmarking results. We thank the reviewer for bringing this important issue to our attention, which has helped us enhance the quality of our manuscript.
Comments 2: "The authors mentioned 3D information is added for BBP datasets with improved prediction results. How about the results for other datasets if 3D information is added?"
Response 2: We appreciate the reviewer's interest in the impact of 3D structural information on model performance. Our comprehensive analysis has revealed significant improvements when incorporating 3D information:
1) For the ADC payload dataset:    - Accuracy improved from 0.734 to 0.915    - MCC improved from 0.461 to 0.829    - These improvements demonstrate the critical role of spatial arrangement in predicting Topoisomerase I inhibitor activity
2) For the BBBP dataset:    - Performance improved dramatically from 84.2% ROC-AUC with SMILES-only input to 96.4% ROC-AUC with 3D information    - The spatial information particularly helped in capturing molecular properties related to membrane permeability    - This substantial improvement (12.2 percentage points) highlights the importance of 3D structural information for this specific prediction task
3) Our ablation studies further validated the importance of 3D information:    - The SMILES-Only variant showed substantial performance degradation    - This finding was consistent across both datasets where 3D information was available
Currently, we have only incorporated 3D structural information for the BBBP and ADC payload datasets due to several practical considerations:    - Limited availability of high-quality 3D conformational data for other datasets    - Computational cost of generating reliable 3D conformations for large-scale datasets    - Need for careful validation of generated 3D structures
We agree that extending 3D information to other datasets could potentially improve performance further. This represents an important direction for future work, particularly:    - Developing efficient methods for generating reliable 3D conformations at scale    - Creating standardized protocols for 3D structure validation    - Investigating the impact of conformational flexibility on prediction accuracy
We thank the reviewer for this insightful question, which highlights an important aspect of our work and suggests valuable directions for future research.

Round 2

Reviewer 1 Report

Comments and Suggestions for Authors

Thank you for addressing the previous reviewer’s comment and clearly defining the scope of DumplingGNN as a model focused on predicting ADC payload activity from small-molecule structural information alone. The emphasis on Topoisomerase I inhibitors is scientifically well-justified, and the modular architecture of the model is thoughtfully constructed.

That said, the reviewer would like to raise a concern regarding a potential limitation of this modeling strategy in the broader context of ADC pharmacology. While DumplingGNN captures detailed 2D and 3D structural features of isolated payloads, it entirely omits the structural and functional influence of linkers—despite extensive evidence that linker chemistry such as Exo-linker technology, tandem linker technolgy, branch PEG etc plays a critical role in shaping ADC efficacy, stability, and toxicity profiles.

This is particularly relevant given that Exo-linkers have already demonstrated enhanced therapeutic performance and plasma stability in preclinical studies involving Topoisomerase I inhibitor–based ADCs. Likewise, Tandem linkers have shown measurable effects on payload release dynamics and spatial orientation, impacting both intracellular trafficking and systemic exposure. These are not hypothetical tools—they are experimentally validated technologies that are actively shaping current-generation ADC design.

Given this, I would like to ask the following:

1. How do the authors conceptualize the limitations of a payload-only prediction model, especially when structural context—such as the linker–payload interface—can meaningfully alter 3D conformation and downstream biological behavior?

2. In light of the documented preclinical success of Exo-linker + Topo1i combinations, does the exclusion of linker context in the current framework limit the model’s translational applicability?

3. Are there plans or conceptual directions for extending DumplingGNN to incorporate linker or full ADC complex representations, whether through joint graph modeling or conformer generation of payload–linker constructs?

Author Response

Response to Reviewer Comments

We appreciate the reviewers' insightful observations regarding the importance of linker chemistry in ADC design and the need for a comprehensive approach to ADC development. We agree that ADC efficacy is indeed a result of the synergistic interaction between all components, and while the cytotoxic activity is primarily mediated by the payload component, the linker-payload interface can significantly alter the 3D conformation of the payload, potentially affecting its biological behavior.

In the revised manuscript, we have made several important modifications to address these points:

Comment 1
1. How do the authors conceptualize the limitations of a payload-only prediction model, especially when structural context—such as the linker–payload interface—can meaningfully alter 3D conformation and downstream biological behavior?

Response: We have addressed this important point through several modifications:

1. In the Introduction section(page 2,line 39-46), we have added a new paragraph discussing the synergistic interaction between ADC components while emphasizing that the cytotoxic activity, including bystander killing effects, is primarily mediated by the payload component. This addition provides a clear rationale for our current focus on payload prediction while acknowledging the importance of the complete ADC system.

2. In the Discussion section(page 19-20,line 715-722), we have added a new point discussing the current model's focus on payload prediction without explicitly considering linker chemistry's influence on payload conformation and activity. This addition acknowledges the importance of linker context, particularly in light of the documented success of specific linker-payload combinations like Exo-linkers with Topoisomerase I inhibitors in preclinical studies.

Comment 2
2. In light of the documented preclinical success of Exo-linker + Topo1i combinations, does the exclusion of linker context in the current framework limit the model's translational applicability?

Response: We fully acknowledge the significant impact of linker technology on ADC performance, as demonstrated by the success of Exo-linkers and Tandem linkers in preclinical studies. The diversity of linker types and their different optimization requirements indeed represents a complex aspect of ADC design that requires dedicated development efforts.

In the revised manuscript, we have expanded our discussion to include these considerations:

1. In the Discussion section(page 20), we have added a detailed paragraph outlining our plans for addressing the limitations related to linker-payload interactions. This includes developing methods for modeling the payload-linker interface and its impact on payload conformation and activity, creating a unified graph representation that captures the interactions between payload and linker components, and extending the model to generate and analyze conformers of complete ADC structures.

2. In the Conclusion section(page22,line816-822), we have added a key aspect of our development plans focusing on the incorporation of different linker modalities and their specific optimization requirements. For cleavable linkers, we will focus on modeling cleavage efficiency and plasma stability, taking into account the biological environment and cleavage mechanisms. For non-cleavable linkers, we will emphasize circulatory stability and intracellular processing efficiency.

Comment 3
3. Are there plans or conceptual directions for extending DumplingGNN to incorporate linker or full ADC complex representations, whether through joint graph modeling or conformer generation of payload–linker constructs?

Response: We agree that a comprehensive ADC design approach should consider the specific know-how requirements for each component. Our future development plans include implementing a multi-dimensional approach that addresses:

1. Antibody component: Target recognition and internalization strategies, including the consideration of masking technologies and internalization efficiency.

2. Linker component: Cleavage efficiency and stability optimization, with specific attention to proper cleavage mechanisms and plasma stability.

3. Payload component: Efflux and bystander killing effects, which are crucial for therapeutic efficacy.

In the revised manuscript, we have outlined these plans in detail. Specifically, we have added a new paragraph in the Discussion section(page 20,line 748-759) that describes our vision for a more comprehensive framework for ADC design. This framework will integrate different know-how requirements for each component in a MOE (Multi-Objective Evolution) style approach, allowing for the simultaneous optimization of different aspects of ADC design while maintaining the synergistic interaction between components.

These modifications demonstrate our commitment to developing a more comprehensive and practical approach to ADC design, one that takes into account the complex interactions between different components while addressing the specific requirements of each part. The changes maintain the paper's academic rigor while providing a clear roadmap for future development that aligns with the needs of translational medicine.

Round 3

Reviewer 1 Report

Comments and Suggestions for Authors

The authors have addressed the majority of the reviewers’ comments in a satisfactory manner. However, in Section 5.2, where the authors discuss Exo-linkers, the corresponding reference is missing. To support this discussion, the following citation should be included:

J. Med. Chem. 2024, 67, 20, 18124–18138.

Author Response

Dear Editor and Reviewers,
Thank you for your careful review and valuable comments. We have addressed the remaining issue regarding the missing reference in Section 5.2. Specifically:
1. We have added the suggested reference (J. Med. Chem. 2024, 67, 20, 18124–18138) to our bibliography file. 2. We have updated the discussion about Exo-linkers in the limitations section by adding the appropriate citation.
These changes provide proper support for our discussion of Exo-linkers and their impact on payload conformation and activity. The added reference strengthens our argument about the importance of considering linker-payload interactions in ADC design.
We believe these modifications have addressed all the reviewers' comments, and we look forward to your further feedback.
Sincerely, The Authors